# Acupuncture and Kinesitherapy Improve Physical Activity More than Kinesitherapy Alone in Patients with Acute Decompensated Chronic Heart Failure with Reduced Ejection Fraction Who Are Already on Optimal Drug Therapy: A Randomized, Sham-Controlled, Double-Blind Clinical Study

**DOI:** 10.3390/biomedicines13010176

**Published:** 2025-01-13

**Authors:** Dejan Ilic, Zoran Jovic, Zorica Mladenovic, Vesna Pejovic, Branislava Lung, Aleksandra Kozic, Slobodan Obradovic

**Affiliations:** 1Clinic for Physical Medicine and Rehabilitation, Military Medical Academy, 11000 Belgrade, Serbia; drdejanilic70@gmail.com (D.I.); pejozok@yahoo.com (V.P.); drbranislavaboskovic@gmail.com (B.L.); 2Clinic for Cardiology, Military Medical Academy, 11000 Belgrade, Serbia; zockydr@gmail.com (Z.J.); zoz3377@gmail.com (Z.M.); 3Medical Faculty of the Military Medical Academy, University of Defense, 11000 Belgrade, Serbia; a.djajic@gmail.com

**Keywords:** acupuncture, kinesitherapy, early rehabilitation, heart failure

## Abstract

**Background/Objectives**: It is unknown how early exercise therapy combined with acupuncture (AP) affects the speed of recovery in patients suffering from heart failure with reduced ejection fraction (HFrEF) who are hospitalized due to worsening HF. **Methods**: A prospective, sham-procedure-controlled, double-blind, randomized clinical trial with three patient groups was conducted. The study included patients with HFrEF who were hospitalized for worsening HF. After initial stabilization, patients were randomly assigned to three groups, the controls (C) treated with optimal drug therapy (ODT); E1 received ODT with kinesitherapy, and sham AP; E2 received ODT, kinesitherapy, and needle AP aiming to enhance compensation. Improvement in physical activity was measured by the modified Barthel index (A test), 2 or 6 min walking tests, an endurance walking test, and decrease of blood brain natriuretic peptide (BNP). **Results**: A total of 120 patients completed the study. The average age was 74 years, 73.3% were male, median EF was 32%, and 83.3% were New York Heart Association (NYHA) Class III. After the fifth day of therapy, the E2 group showed the greatest improvement in the A test compared to other groups (*p* = 0.022). After 5 days, 80%, 50%, and 10% achieved an A test score of 30 or higher, demonstrating functional independence, (*p* < 0.01) in the E2, E1, and C groups, respectively. At hospital discharge, the E2 group showed significantly better walking endurance compared to the E1 and C groups (*p* < 0.001). At discharge, BNP decreased significantly more in both E groups, compared to the C group. **Conclusions**: Short-term early rehabilitation with AP and kinesitherapy significantly improves the functional status of HFrEF decompensated patients when compared to kinesitherapy alone and ODT.

## 1. Introduction

Heart failure (HF), a condition in which the heart is unable to deliver a sufficient amount of blood to peripheral tissues and meet oxygen requirements while maintaining venous inflow, is the leading cause of death in developed countries (North America and Europe) [1,2]. More than 64 million people suffer from HF, half of which have a reduced left ventricular ejection fraction (LVEF). The prevalence in the general population is 1–3%. The 1-year mortality rate is 15–30% [1]. The clinical presentation of patients with HF is characterized by dyspnea, orthopnea, fatigue, and peripheral oedema. They very often suffer from depression and poor quality of life and are often re-hospitalized, and the rehabilitation of such patients has not been well studied [3,4,5]. The aim of cardiac rehabilitation (CR) is to improve the function of the cardiovascular system (CVS) [6]. CR is carried out using exercise therapy according to the patient’s capabilities. Programs are administered individually according to intensity, duration, and frequency. Cardiac rehabilitation has evolved significantly since 1995, when the U.S. Agency for Health Care Policy and Research (AHCPR) published the Clinical Practice Guideline on Cardiac Rehabilitation, emphasizing its role in managing coronary heart disease and heart failure. Defined as a multifactorial process including exercise training, education, and lifestyle counseling, it has advanced over time [7]. The 2022 clinical practice guidelines from the American College of Cardiology (ACC) and the American Heart Association (AHA) elevated cardiac rehabilitation to a Class 1 A level recommendation, endorsing supervised exercise training for patients with stable heart failure with reduced ejection fraction to improve functional status, exercise performance, and quality of life [8]. According to the American College of Sports Medicine’s (ACSM) guidelines for cardiac rehabilitation, the usual treatment consists of short walks with a progressive increase in the distance 3–4 times per day and upper extremity exercises; however, the optimal timing and duration of therapy are unknown [9]. The Early Rehabilitation in Cardiology-Heart Failure (ERIC-HF) approach is safe and feasible [6], with exercises lasting 5–20 min each day, 5 days a week, and is frequently used in the early rehabilitation of hospitalized patients and research. Our rehabilitation program is similar: we start with breathing exercises and peripheral circulation exercises. Then, exercises for the upper and lower extremities are introduced. Gradual verticalization begins in bed, progressing to standing and walking with a therapist or assistance.

In their study, Kitzman et al. showed that three months of rehabilitation significantly improved physical fitness and quality of life, and reduced depression in elderly HF patients [5]. The German Society of Cardiology and the German Association for Preventive Rehabilitation created an exercise program for patients with heart failure. The research involved 12 patients with LVEF < 45%, New York Heart Association (NYHA) functional class II and III, who engaged in a weekly, 1-h supervised exercise with a physician for one year. Subsequent checks at 4, 8, and 12 months showed a significant rise in LVEF, improvement in quality of life, decrease in total cholesterol, and decrease in BNP levels. The rise in exercise capability and the 6 min walk test (6MWT) both displayed upward trends; however, they were not statistically significant. In the rehabilitation process, there were no negative cardiovascular (CV) incidents [10]. Delgado et al. investigated the effects of an aerobic exercise program (kinesitherapy) on patients with decompensated HF during early rehabilitation. Their study involved 257 patients with reduced LVEF and NYHA III and IV functional class. While hospitalized, patients participated in a regimen of aerobic activities, including breathing exercises, walking, using an ergo bike (a stationary exercise bike that measures work output), and climbing stairs, twice a day, five times a week, tailored to their individual capacities. The study demonstrated a significant increase in walking distance, as measured by the 6MWT, and an improvement in functional independence, as assessed by the Barthel index (a commonly used tool for measuring a patient’s ability to perform basic daily activities independently, such as feeding, bathing, dressing, mobility, and toileting [11]. In a meta-analysis, Chinese researchers reviewed six randomized controlled trials involving 668 patients with acute decompensated HF (LVEF < 40%) who underwent early rehabilitation, sourced from the PubMed, CENTRAL, EMBASE, and WANFANG databases up to July 2022. They found a significant improvement in exercise capacity, as indicated by higher scores in the 6 min walk test (6MWT) and the Short Physical Performance Battery (SPPB) in the intervention group compared to the control group. Additionally, the rate of readmission was significantly lower in individuals who received early rehabilitation [12]. Previous studies have confirmed that acupuncture (AP) has an analgesic and immunomodulating impact by activating the nervous system [13,14]. AP has been shown to have sympatholytic, vasodilatory, and cardioprotective properties, potentially enhancing heart function in HF patients [15,16]. A meta-analysis conducted in October 2019 by Chinese researchers, which involved 32 randomized controlled trials with a total of 2499 patients, outlined the beneficial effects of AP in HF. They contrasted patients who received regular drug treatment with those who received AP alongside drugs. In one study, patients who were given AP showed a significant improvement in LVEF, minute volume, and 6MWT along with a decrease in BNP levels [17]. BNP is important in predicting and diagnosing heart failure [18]. There is limited research on the combined effects of kinesitherapy and acupuncture in early rehabilitation for HF patients with cardiac decompensation. In this study, we aim to examine how the combined impact of AP and exercise compares to the impact achieved through exercise alone.

## 2. Materials and Methods

The intended study is a prospective, double-blinded, randomized clinical trial with two experimental groups and one control group, using a placebo for comparison. Patients were followed up for 5–10 days, with the study lasting from January 2022 to May 2024. The study was carried out at the Cardiology Clinic of the Military Medical Academy (MMA), Belgrade, Serbia.

Eligibility criteria:

Patients admitted for decompensated HF were chosen based on clinical presentation, medical history, and echocardiographic findings within the first 72 h. They were diagnosed with chronic heart failure with reduced left ventricular ejection fraction (LVEF < 40%) and NYHA II or III functional classes [19,20]. The minimum LVEF for the research was 15%. Following stabilization of hemodynamics and successful response to diuretics, patients were sorted into 3 groups within 72 h of being admitted to the hospital.

Control group C of participants: Participants received standard drug treatment. The optimal therapy, which had a class I recommendation as of 2021, includes four groups of medications: beta-blockers, ACE inhibitors or angiotensin receptor antagonists or valsartan-sacubitril, mineralocorticoid receptor antagonists, and sodium/glucose cotransporter (SGLT) 2 receptor inhibitors [21,22].

Experimental patient group E1: In addition to optimal drug therapy, this group received a standardized daily kinesitherapy program tailored to each individual, similar to that used in early rehabilitation for CV patients following cardiac procedures, acute myocardial infarction, and other heart-related diseases. Kinesitherapy is tailored to each individual, lasting between 5 and 10 days, based on the patient’s health status. Patients completed the exercises while lying down, sitting, or standing based on their overall health and capacity to participate in the activities outlined in the A test. Breathing techniques, exercises for circulation in the extremities, and exercises for the arms and legs were utilized. Patients with compensated heart failure began kinesitherapy (KTH) within 72 h of admission, while those with decompensated heart failure started rehabilitation once they became compensated. Patients with decompensated heart failure were defined as those experiencing acute or progressive hemodynamic instability, and acute or chronic worsening of symptoms such as pulmonary congestion, peripheral edema, and dyspnea, often requiring urgent medical intervention. Patients without these symptoms were defined as compensated. In the E1 experimental group, patients were given drug therapy and kinesitherapy in addition to placebo acupuncture. The acupuncture placebo involved placing tubes or guides close to, but not directly on, the acupuncture point or channel. The tubes were attached by using double-sided adhesive patches, with shorter acupuncture needles inserted into them to prevent piercing the skin [23].

Experimental patient group E2: In addition to optimal drug therapy, this group followed a kinesitherapy program based on the same principles as the E1 group combined with classic AP. In our study, we utilized AP points which are commonly applied in the care of CV patients as per classical theory and prior studies. Neiguan (PC 6), Shenmen (HT 7), Hegu (LI 4), Shanzhong (RN 17), Yinlingquan (Sp 9) on both sides, Zusanli (ST 36), and Baihui (Du 20) are acupuncture points used to boost heart function, enhance urination, and lower stress [24]. An experienced acupuncturist-physiatrist applied AP; they had graduated from the School of Traditional Chinese Medicine (TCM) at the II Military University in Shanghai, China, in the academic year 2006/07. The needles were left in place for 20 min with no extra stimulation. Fresh sterile acupuncture needles were utilized. The needles, from the Chinese brand “Hwato”, had measurements of 30 mm in length and 0.3 mm in thickness. Every individual patient had a collection of needles. The AP took place within 5–10 business (working) days.

Exclusion criteria for the study:

Patients were excluded if they had been admitted to the hospital within the last month due to a heart attack and/or myocardial revascularization, were scheduled for heart surgery, diagnosed with malignant disease, immobile (excluding certain conditions), anemic with hemoglobin levels below 90 g/L, had other comorbidities hindering physical therapy, uncooperative, or exhibited a Mini-Mental State score below 24 (which was assessed by a speech therapist specializing in psychological testing). Further exclusion criteria included abnormal vital signs: blood pressure above 180/120 mmHg and below 90/60 mmHg, heart rate of 130/min or higher, ventricular tachycardia or fibrillation within 72 h and fever.

The research examined the following dependent variables:

1. Parameters for cardiological monitoring included blood pressure, heart rate, and BNP (natriuretic peptide type B). BNP was measured in venous blood on the first day (before the start of therapy) and on the last day (1 h after the last treatment). 2. Physiatric monitoring factors: The A-test monitors and evaluates 10 activities, including those performed while in bed, standing, moving, and walking. The ratings for each task range from 0 (not fully accomplished) to 5 (completely autonomous and secure), with a total possible score of 50. A higher total score indicates a higher level of functionality for the patient. This assessment covers all early rehabilitation activities, including moving in bed, sitting on the bed, using the restroom, standing, and walking with assistance or unassisted. It can be used to assess functional ability, in the event of any disability caused by illness or injury [25].

The 2 min walking test (2MWT) and 6 min walking test (6MWT) assessed the examinee’s strength and endurance. This examination was performed on each patient at both the start and conclusion of the follow-up period, which corresponds to the therapy phase. The number of meters walked was calculated either within two minutes or within six minutes, based on the patient’s level of endurance. If the patient could not finish the 2MWT, the score for walking distance was recorded in the walking endurance section of Test A. A score of 0 indicates the patient is unable to walk, while a score of 1 means they can walk up to 5 m. A score of 2 indicates walking up to 15 m, a score of 3 up to 50 m, a score of 4 up to 100 m, and a score of 5 over 100 m [26,27].

### 2.1. Sample Size

The size measurement for three separate samples was determined with a 6% margin of error and an 80% confidence level, for the distribution of individuals among the groups in a 1:1:1 ratio. There were at least 114 respondents in total, split evenly into three groups. All groups consisted of at least 38 patients each, which satisfies the requirement for large statistical samples mentioned by Kovačević-Kostić (2015) [28]. To confirm the data, the study sample size was determined using the Raosoft online calculator (http://www.raosoft.com/samplesize.html (accessed on 15 October 2024); Raosoft, Seattle, WA, USA) [29]. Additional patients were included in case some patients did not meet the criteria for research inclusion. The formula used to determine the adjusted sample size was n1 = n/(1 − e), where n is the original sample size, n1 is the adjusted sample size, and e is equal to 10% of the original sample size (Sakpal, 2010) [30]. The revised sample size was N = 127. This paper will detail all the factors leading to attrition of the research (fatal consequences, deterioration of the primary condition, additional issues like COVID-19, personal and familial motives for discontinuation, etc.), categorized by patient groups.

### 2.2. Statistical Analysis

Frequencies and percentages are used to describe categorical variables. The Shapiro–Wilk test was utilized to assess the normality of the distribution of numerical variables. Because the variables did not satisfy the normality criterion, the Median and Interquartile ranges were utilized. The Kruskal–Wallis test and the Mann–Whitney U test were used to compare differences among groups. The differences between two or more repeated measurements were tested using the Wilcoxon signed-ranks test and the Friedman test, as well as Kendall’s W. The Chi-square test was used to analyze the association between categorical variables. We used a combined analysis of variance (SPANOVA) to see if the change in test outcomes measured at admission and discharge varied by respondent group. All statistical analyses were performed using IBM Statistics SPSS version 24.0 (IBM, Armonk, NY, USA).

The Ethical Committee of the Military Medical Academy Belgrade approved the study at its meeting on 1 December 2022 (decision number 56/2022), and all patients provided their written consent for participation.

## 3. Results

A total of 280 patients were assessed for eligibility. Among them, 139 patients did not meet the criteria to enter the study, primarily due to cardiovascular diseases (56.83%), followed by poor general conditions related to other internal medicine conditions (7.19%), infectious syndromes (9.35%), poor patient cooperation (12.94%), and other reasons (13.67%). Additionally, 21 patients dropped out during the study. The study flow chart is presented in Figure 1.

A total of 120 participants were involved in the study; among them, 40 patients were on medication therapy as well as adapted kinesitherapy and acupuncture. Respondents had an average age of 74.0 years, with an interquartile range of 11.7 years, and 73.3% were men. Arterial hypertension was the most prevalent coexisting medical condition (70.0%). The median LVEF value was 32.0, with an interquartile range of 9.5, while 83.3% of patients were in the NYHA III stage. All the mentioned parameters show that the three groups were similar in terms of uniformity (Table 1).

A comparison between the three groups in the functional status measured by the A test during 10 days is presented in Table 2. There was a significant improvement in the functional status of all groups on the 5th and 10th days of measurement (*p* < 0.001). The E2 group showed the largest A test score increase in the percentage, both on the 5th and 10th day, compared to the other two groups. Initially, the three groups were all on an equal playing field in terms of their performance on the test, meaning there were no intergroup disparities. On the 5th day, significant variations were observed between groups E1 and E2 (*p* = 0.022), with group E2 displaying superior outcomes (Median = 35.5 (IQR = 14.0)) in comparison to the E1 group (Median = 30.5 (IQR = 10.5)). Both groups surpassed the C group by the 5th day, with a significance level of *p* < 0.001. By day 10, E1 (Median = 35.0 (IQR = 12.0)) and E2 (Median = 41.0 (IQR = 13.0)) groups showed superior results in the A test compared to the C group (Me = 25.0 (IQR = 11.2)), with *p* values of 0.045 and 0.005, respectively. The E1 and E2 groups had comparable functional status on the 10th day.

After five days, a higher percentage of patients in the E2 group achieved an A test score of 30 or above, compared to the E1 and C groups (80% vs. 50% vs. 10%, respectively; *p* < 0.01), indicating functional independence, the ability to walk, and eligibility for hospital discharge.

A comparison between the three groups in the 2MWT, 6MWT, and endurance walking test between admission and discharge, along with within-group comparisons, is presented in Table 3.

There was no significant improvement in 2MWT between admission and discharge within the C group. The E1 group improved from a distance of median = 7.62 m (IQR = 6.0) at admission to a median = 50.0 m (IQR = 85.0) at discharge (*p* < 0.001). The E2 group increased from a distance of median = 5.0 m (IQR = 0.0) at admission to a median = 120.0 m (IQR = 63.0) at discharge (*p* = 0.001). The distance walked within 2 min at discharge differed significantly between the E1 and E2 groups (*p* ˂ 0.001), indicating that the E2 group performed better. Both experimental groups performed better in 2MWT at discharge than the control group (*p* < 0.001). There was no difference between the three groups in the 2MWT at admission.

In 6MWT, the E1 group improved from a distance of median = 90.0 m (IQR = 20.0) at admission to a median = 190.0 m (IQR = 85.0) at discharge (*p* = 0.042). The E2 group improved from a distance of median = 80.0 m (IQR = 75.0) at admission to a median = 310.0 m (IQR = 40.0) at discharge (*p* = 0.042). Patients in the E2 group covered longer distances in the 6MWT compared to those in the E1 group, both at admission (*p* = 0.033) and at discharge (*p* = 0.016).

Upon admission to the hospital, all groups showed similar endurance walking test results. The endurance walking test demonstrated improvement in all groups from admission to discharge (*p* < 0.001). After discharge, both experimental groups demonstrated significantly better results in the endurance walking test compared to the control group, with the E2 group also showing significantly better results than the E1 group (*p* < 0.001).

The SPANOVA test showed that the combination of KTH and AP in the E2 group improved functionality at discharge the most, compared to the other two groups, as assessed by the A test (Wilks’ lambda = 0.035, F = 4.840, *p* < 0.001, η^2^ = 0.813), Figure 2. Also, KTH and AP combined resulted in a significantly greater improvement in the walking endurance test compared to the other two groups (Wilks’ lambda = 0.496, F = 59.50, *p* < 0.001, η^2^ = 0.504), Figure 3. All squared eta (η^2^) values are high, indicating that the kind of therapy (KTH and AP) has a strong combined effect on the change in results when comparing the values at admission and discharge.

In the E2 group, blood concentrations of BNP decreased more pronouncedly from admission to discharge, compared to the E1 and C groups (64.1%, *p* < 0.001; 46.0%, *p* < 0.001; 12.9%, *p* = 0.001; respectively). The groups had the same average BNP upon admission. There were variances in the BNP delta discrepancy in measurements between C vs. E1 (*p* = 0.001) and C vs. E2 (*p* < 0.001). There was no difference in the BNP delta discrepancy between the two experimental groups (Figure 4).

Differences between the three groups in the duration of hospitalization and follow-up are presented in Table 4. There was no difference in the average number of hospital days among the three groups. Similarly, there was no significant difference among the groups in the number of days required for monitoring and rehabilitation. Patients in the control group spent an average of 9.00 days in the hospital, compared to 8.00 days in group E1 and 10.00 days in group E2. The mean follow-up days in the C group and the rehabilitation days in the experimental groups are equal to 6.00.

The C group had the highest composite rate of all-cause death and rehospitalization due to worsening heart failure, reaching 27.5% six months after discharge, followed by the E1 group at 17.5% and the E2 group at 10%. Mortality rates showed no significant differences among the groups (Table 5).

## 4. Discussion

Our study is the first prospective, placebo-controlled, double-blind, randomized clinical trial in patients with HF and reduced LVEF after decompensation, demonstrating the combined effects of AP and KTH in early rehabilitation. Patients receiving combined therapy (AP and KTH) alongside pharmacological treatment exhibited the fastest functional recovery after 5 days of rehabilitation, as shown by improvements in the A test, 2 min and 6 min walk tests, and walking endurance test. The E2 group, which received both AP and KTH, achieved functional independence earlier than the other groups, highlighting the additive effect of AP on KTH, which serves as the foundation of rehabilitation.

Functional recovery was evaluated using the Barthel index, a tool for assessing functional independence in daily activities such as eating, dressing, and walking. The Barthel index is commonly used in acute heart diseases [31] and cardiac rehabilitation [32], and it is a predictor of mortality in HF. A Barthel index score below 85 at discharge is associated with increased mortality [33]. While the Barthel index is effective for broad evaluations, prior research revealed that the A test is often superior for detecting daily changes in functional status [25]. Originally designed for trauma and orthopedic patients, the A test has been adapted for cardiac rehabilitation since 2012. It identifies patients who are progressing or lagging in functional recovery during therapy [26]. Patients should be assessed based on the type and severity of their functional disability rather than solely on their diagnosis [27,34,35].

By day 4, patients in the E2 group had achieved the same functional capacity (A = 30.5) that the E1 group reached on day 5, indicating earlier readiness for discharge (Table 2). This represents a significant improvement, potentially reducing treatment costs and hospital-acquired infections. After 5 days, the E2 group reached optimal functional recovery (A = 35.5). From day 5 to day 10, functional improvement continued across all groups, albeit at a slower rate. By day 10, the E2 and E1 groups had comparable functional status, while the C group lagged significantly. The C group’s A test score increased from 22 on day 5 to 25 on day 10, indicating much slower recovery and uncertain full recovery. These findings emphasize the importance of early rehabilitation for all HF patients, ideally combining exercises with AP where feasible.

Achieving functional independence by day 5 was critical. Patients with A test scores above 30 were considered functionally ready for discharge. In the E2 group, 80% of patients (32/40) could independently get out of bed, use the restroom, and walk. In the E1 group, 50% of patients met these criteria, compared to only 10% in the C group.

The walking endurance test, a component of the A test, showed that at discharge, the E2 group outperformed the E1 group, with an average walking endurance grade of 4 (able to walk up to 100 m) compared to grade 3 in the E1 group (able to walk up to 50 m). The C group achieved a grade of 1.5 (5–15 m), reflecting significant but limited improvement. At admission, most patients in all groups were bedridden or could only walk up to 5 m within their rooms (grades 0–1).

The 2MWT is well suited for assessing functional capacity in geriatric patients and those recovering from heart surgery or acute myocardial infarction [36]. However, it is less sensitive and correlates less strongly with daily living activities compared to the 6MWT [37]. The 2MWT is particularly useful for NYHA class III and IV patients [38]. The average distance covered by healthy individuals aged 70–74 in the 2MWT is 172.2 m for men and 145.9 m for women [39]. Similarly, 6MWT distances vary by age and gender, ranging from 400 to 700 m [40], with men walking 30 m farther than women on average [41].

The E1 group achieved 60 m more in 6MWT at discharge compared to the C group (Table 3), which is similar to the results obtained in the 2022 study by Delgado et al. [11], where the difference between the experimental group that exercises and the control group is 59 m. However, in our study, no significant difference was observed between the E1 and C groups in the 6MWT, likely due to the small sample size of the C group, which consisted of only two patients. In a study by Dr. Johannes Beck and colleagues at the University Hospital in Heidelberg, 17 subjects undergoing classical AP demonstrated a 32-m improvement in the 6MWT compared to the placebo AP group [42]. In our study, the difference between the groups receiving classic AP and KTH and the placebo AP and KTH was 120 m, a significant result.

The 6MWT was the most demanding test, with only 12 patients completing it upon admission—five from each of the E1 and E2 groups, and just two from the C group. In contrast, 46 patients completed the 2MWT (14 from the E1 group, 13 from the E2 group, and 19 from the C group), while 62 patients were unable to walk upon admission. The endurance walking test was the most comprehensive, administered to all 120 patients at both admission and discharge. A potential modification to improve its accuracy for patients with severe clinical conditions who cannot walk upon hospital admission could involve measuring the distance walked without a time limit for small distances (up to 100 m), avoiding the graded intervals used in the current method.

The meta-analysis mentioned in the introduction reported a significant increase in the 6MWT (MD = 43.6, 95% CI = [37.43, 49.77], I2 = 0%, *p* < 0.0001) and a reduction in BNP (MD = −227.99, 95% CI = [−337.30, −118.68], I2 = 96%, *p* < 0.0001) between patients treated with acupuncture and those receiving only medical therapy [17]. In a separate meta-analysis of 565 patients with HF, Smart et al. found that kinesitherapy reduced BNP levels by 28.3%, with the greatest reductions observed in patients with an LVEF < 34% [43]. Our study, with a mean EF of 32%, found that the E2 group exhibited the highest BNP reduction compared to the E1 and C groups. (Figure 4).

In a meta-analysis of acute heart failure studies conducted by Chinese authors, acupuncture reduced intensive care unit (ICU) stay by 2.2 days (95% CI 1.26, 3.14) and lowered the probability of rehospitalization to 0.53 (95% CI 0.28, 0.99) [44]. While our study did not demonstrate a reduction in hospital stays for the experimental groups (Table 4), we did observe a potential reduction in hospitalization by one day in the E2 group compared to the E1 group. Reews et al. found that six months post-discharge, patients who had undergone exercise therapy had 29% fewer rehospitalizations than the control group [45]. Similarly, Zhong’s study reported a lower rehospitalization rate within six months for the exercise group (12.5%) compared to the control group (23.6%) [46]. Consistent with these findings, our study showed a reduction in rehospitalization rates six months post-discharge: 5% in the E2 group, significantly lower than the 12.5% in the E1 group and the 27.5% in the C group (Table 5). After 6 months of rehabilitation in elderly patients with HF, Kitzman et al. reported nearly twice as many fatalities in the experimental group (15 CV deaths out of 175 patients) compared to 8 CV deaths out of 174 patients in the control group. The total death rate is 6.59% [5]. In our study, the overall number of deaths after 6 months was 5.83% (7/120), as shown in Table 5.

Acupuncture demonstrates excellent cost-effectiveness as an adjunct therapy, offering substantial benefits at relatively low costs. The primary material requirements are minimal, including small spaces, beds, chairs, and inexpensive disposable acupuncture needles. While this makes acupuncture accessible, the greatest logistical challenges are the availability of trained professionals and potential resistance to incorporating acupuncture into conventional medical frameworks. Future studies are needed to quantify the long-term cost savings of acupuncture, such as reduced medication use, fewer rehospitalizations, and improved functional recovery, particularly in patients with chronic diseases like heart failure. These studies would provide robust evidence for the broader adoption of acupuncture into standard care.

### Study Limitation

The results of our study should be interpreted with caution due to the small sample size, which may limit the generalizability of the findings. Future studies with larger, more diverse cohorts are essential to understand long-term outcomes better and enhance generalizability. These studies should evaluate patients undergoing different rehabilitation approaches, including secondary rehabilitation (e.g., spa-based programs) and rehabilitation in home settings. Key outcomes to assess would include one-year functional recovery, rehospitalization rates, mortality, and treatment costs.

Another limitation of this study is the relatively short rehabilitation period, which was limited to 5–10 working days for all patients, including those in the control group. Although the study was designed for a 10-day rehabilitation period for all groups, for objective reasons, patients who could go home earlier were discharged. This allowed us to draw additional conclusions. Specifically, 80% of patients receiving combined acupuncture (AP) and kinesitherapy (KTH) achieved functional independence and were ready for discharge after 5 days, compared to 50% in the kinesitherapy-only group and only 10% in the control group.

The decision to shorten the rehabilitation period was influenced by the constraints of the COVID-19 pandemic. Given the circumstances, we opted for a brief early rehabilitation period. Additionally, during the study, six patients tested positive for COVID-19. Two were diagnosed upon admission and did not participate in the trial, while four were diagnosed during hospitalization and were excluded from the study, falling into the dropout group. All patients with confirmed COVID-19 infection were transferred to the designated COVID-19 hospital for appropriate care.

Although the placebo effects of acupuncture have been documented in other studies—arising from factors such as patient expectations and the ritual of acupuncture [47]—our study minimized these effects by focusing on objective measures such as functional recovery (A test), walking endurance (2MWT, 6MWT), and biochemical parameters (BNP levels). These measures provided reliable and quantifiable data, reducing the potential influence of placebo effects. However, future studies could include additional controls to further explore this aspect.

## 5. Conclusions

Early kinesitherapy accelerates the functional recovery of patients hospitalized due to decompensated heart failure with reduced ejection fraction. Acupuncture synergizes with kinesitherapy, and patients undergoing the combination of these two procedures achieve the best early functional recovery. Further research is needed to determine whether the application of combined therapy affects the long-term stability of patients after discharge.

## Figures and Tables

**Figure 1 biomedicines-13-00176-f001:**
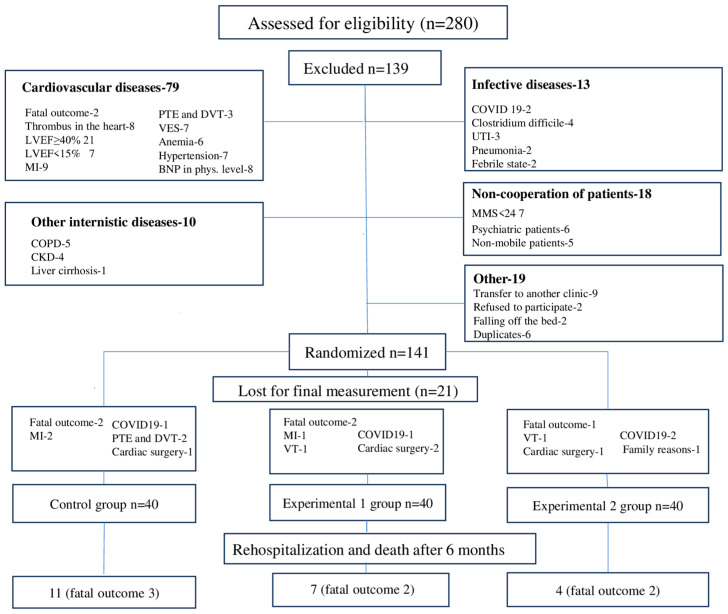
Flowchart of the study. BNP—brain natriuretic peptide, CKD—chronic kidney disease, COPD—chronic obstructive pulmonary disease, DVT—deep vein thrombosis, LVEF—left ventricle ejection fraction, MI—myocardial infarction, MMS—mini-mental score, PTE—pulmonary thromboembolism, UTI—urinary tract infections, VES—ventricular extrasystoles, VT—ventricular tachycardia.

**Figure 2 biomedicines-13-00176-f002:**
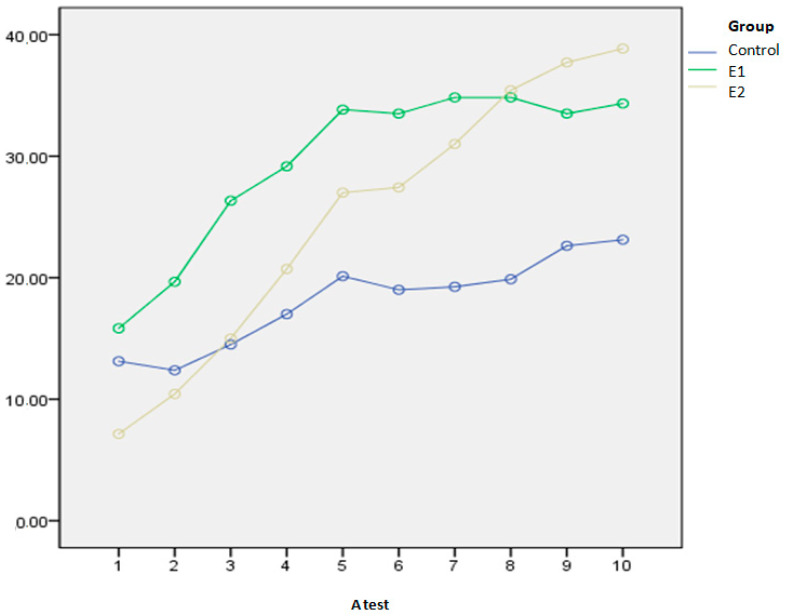
The mean values of the A test during the study in three examined groups.

**Figure 3 biomedicines-13-00176-f003:**
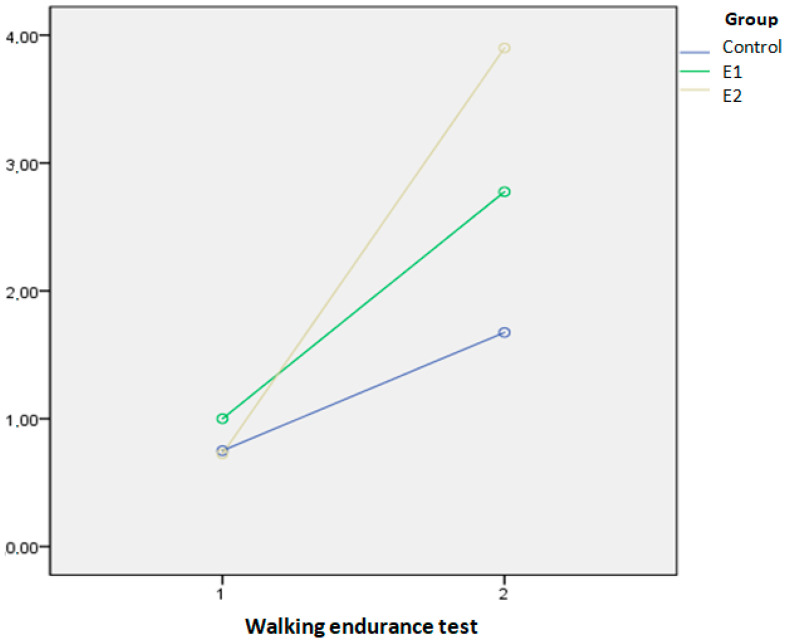
The median values of the walking endurance test at admission and discharge in three examined groups.

**Figure 4 biomedicines-13-00176-f004:**
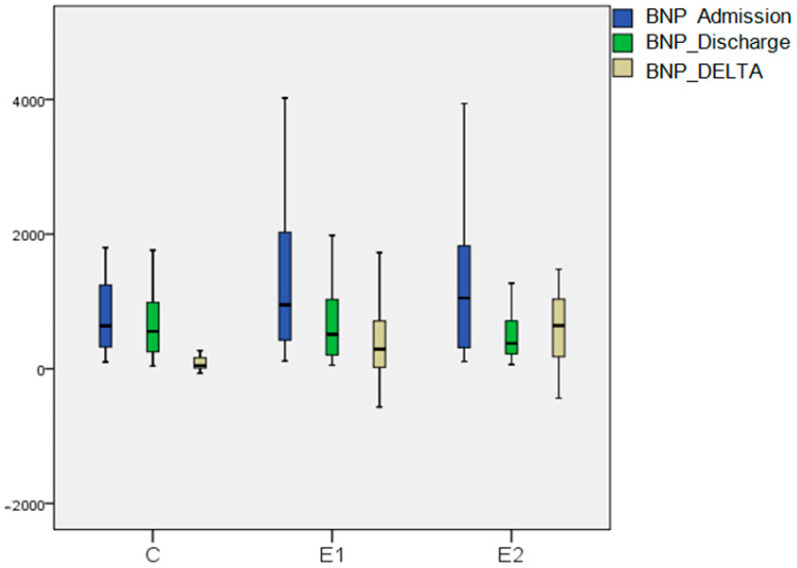
Box Plot of BNP at admission and discharge in 3 examined groups.

**Table 1 biomedicines-13-00176-t001:** General and clinical characteristics of patients.

General Characteristics of the Respondents		Patients: Grouped by Treatment			*p* Value
	Total	C	E1	E2	
	N = 120	N = 40	N = 40	N = 40	
Gender:					0.958 ^a^
Male	88 (73.3%)	29 (72.5%)	29 (72.5%)	30 (75%)	
Female	32 (26.7%)	11 (27.5%)	11 (27.5%)	10 (25%)	
Age (years)	74.0 (11.7), 30–91	73.5 (13.7), 49–88	76.0 (12.7), 30–88	73.5 (10.5), 57–91	0.505 ^b^
Comorbidities:					
Atrial fibrillation	52 (43.3%)	14 (35.0%)	19 (47.5%)	19 (47.5%)	0.428 ^a^
Hypertension	84 (70.0%)	27 (67.5%)	29 (72.5%)	28 (70.0%)	0.888 ^a^
Diabetes mellitus	51 (42.5%)	15 (37.5%)	15 (37.5%)	21 (52.5%)	0.293 ^a^
COVID-19	31 (25.8%)	11 (27.5%)	11 (27.5%)	9 (22.5%)	0.840 ^a^
Smoker	66 (55.0%)	22 (55.0%)	22 (55.0%)	22 (55.0%)	1.000 ^a^
FH of CVD	76 (63.3%)	24 (60.0%)	24 (60.0%)	28 (70.0%)	0.563 ^a^
NYHA scale					
II	20 (16.7%)	7 (17.5%)	6 (15.0%)	7 (17.5%)	0.942 ^a^
III	100 (83.3%)	33 (82.5%)	34 (85.0%)	33 (82.5%)	
LVEF	32.0 (9.5), 15–39	35.0 (10.7), 15–39	35.0 (11.7), 15–39	31.5 (91.5), 18–39	0.277 ^b^

^a^ Chi-square test. ^b^ Kruskal–Wallis Test. Numeric data is shown as Median (Interquartile Range), Minimum–Maximum. E1—experimental group 1. E2—experimental group 2. C—control group. LVEF—left ventricular ejection fraction. FH of CVD—family history of cardiovascular diseases.

**Table 2 biomedicines-13-00176-t002:** A test during 10 days in 3 examined groups.

	All	C	E1	E2	C vs. E1	C vs. E2	E1 vs. E2
	Me (IQR)	Me (IQR)	Me (IQR)	Me (IQR)	*p* Value ^b^	*p* Value ^b^	*p* Value ^b^
A test							
1st day	8.0 (14.0)	9.0 (11.7)	8.0 (15.5)	7.0 (5.0)	0.681	0.211	0.436
2nd day	14.0 (14.5)	10.0 (11.7)	14.0 (16.0)	15.5 (13.0)	0.019	0.042	0.847
3rd day	19.5 (17.0)	14.5 (14.5)	21.0 (14.5)	23.0 (17.0)	0.000	0.000	0.651
4th day	25.0 (17.5)	19.0 (12.7)	27.0 (10.2)	30.5 (20.0)	0.000	0.000	0.198
5th day	30.0 (14.5)	22.0 (14.0)	30.5 (10.5)	35.5 (14.0)	0.000	0.000	0.022
% of change	212.5%	111.1%	137.5%	407.1%			
*p* Value ^a^	<0.001	<0.001	<0.001	<0.001			
*p* Value ^c^	<0.001	<0.001	<0.001	<0.001			
6th day	30.5 (18.5)	22.0 (14.7)	32.0 (13.5)	37.5 (15.0)	0.000	0.000	0.113
7th day	32.0 (18.0)	20.0 (7.0)	33.0 (10.7)	37.0 (17.0)	0.000	0.000	0.393
8th day	33.5 (16.0)	21.0 (7.5)	35.0 (13.0)	35.0 (13.0)	0.001	0.000	0.662
9th day	33.0 (16.5)	22.0 (11.2)	34.5 (14.0)	38.0 (12.0)	0.034	0.001	0.355
10th day	32.0 (15.5)	25.0 (11.2)	35.0 (12.0)	41.0 (13.0)	0.045	0.005	0.352
% of change	300.0%	177.7%	337.5%	485.7%			
*p* Value ^a^	<0.001	<0.001	<0.001	<0.001			
*p* Value ^c^	<0.001	<0.001	<0.001	<0.001			

^a^ Friedman test. ^b^ Mann–Whitney U test. ^c^ Kendall’s W. Me (IQR) = Median (Interquartile Range).

**Table 3 biomedicines-13-00176-t003:** 2MWT, 6MWT, and endurance walking test at admission and discharge in 3 examined groups.

	All	C	E1	E2	C vs. E1	C vs. E2	E1 vs. E2
	Me (IQR)	Me (IQR)	Me (IQR)	Me (IQR)	*p* Value ^b^	*p* Value ^b^	*p* Value ^b^
* 2MWT							
Admission	5.0 (0.0)	5.0 (0.0)	7.62 (6.0)	5.0 (0.0)	0.206	0.879	0.395
Discharge	32.5 (99.0)	5.0 (5.0)	50.0 (85.0)	120.0 (63.0)	<0.001	<0.001	<0.001
*p* Value ^a^	<0.001	0.083	0.001	0.001			
* 6MWT							
Admission	80.0 (10.0)	95.00 (25.0)	90.00 (20.0)	80.0 (75.0)	0.417	0.042	0.033
Discharge	240.00 (153.0)	130.00 (30.0)	190.00 (85.0)	310.0 (40.0)	0.121	0.049	0.016
*p* Value ^a^	0.002	0.180	0.042	0.042			
Endurance walking test							
Admission	0.0 (1.0)	1.0 (1.0)	0.0 (1.7)	0.0 (1.0)	0.826	0.379	0.354
Discharge	3.0 (1.0)	1.5 (1.0)	3.0 (1.7)	4.0 (2.0)	<0.001	<0.001	<0.001
*p* Value ^a^	<0.001	<0.001	<0.001	<0.001			

^a^ Wilcoxon signed-rank test. ^b^ Mann–Whitney U test. Me (IQR) = Median (Interquartile Range). * Missing values N (%) for E1, E2, and C group, respectively: 2MWT—26 (65.0), 27 (67.5), 21 (52.5); 6MWT—35 (87.5), 35 (87.5), 38 (95.0).

**Table 4 biomedicines-13-00176-t004:** Duration of hospitalization and follow-up: intergroup differences.

		All Patients	Control	E1	E2	H	df	*p*
N^o^ of hospital days	Min	5.00	5.00	5.00	6.00	2.203	2	0.332
	Max	28.00	21.00	23.00	28.00			
	Me	9.00	9.00	8.0	10.00			
	M	10.49	10.13	10.25	11.10			
	SD	4.21	3.64	4.58	4.38			
N^o^ of days followed	Min	5.00	5.00	5.00	5.00	0.490	2	0.783
	Max	10.00	10.00	10.00	10.00			
	Me	6.00	6.00	6.00	6.00			
	M	6.59	6.63	6.43	6.73			
	SD	1.90	1.97	1.82	1.95			

Min = minimal sample value, Max = maximal sample value, M = arithmetic mean, SD = standard deviation, Me = median, H = Kruskal–Wallis test, df = degree of freedom, *p* = statistical significance.

**Table 5 biomedicines-13-00176-t005:** Rehospitalization 6 months after discharge.

6 Months After Discharge	All Patients, N (%)	C, N (%)	E1, N (%)	E2, N (%)
Rehospitalization + exitus	22 (18.33)	11 (27.5)	7 (17.5)	4 (10)
Rehospitalization	15 (12.5)	8 (20)	5 (12.5)	2 (5)
Fatal outcome	7 (5.83)	3 (7.5)	2 (5)	2 (5)

## Data Availability

The data generated in the present study may be requested from the corresponding author.

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
