# Peer review of "Acupuncture and Kinesitherapy Improve Physical Activity More than Kinesitherapy Alone in Patients with Acute Decompensated Chronic Heart Failure with Reduced Ejection Fraction Who Are Already on Optimal Drug Therapy: A Randomized, Sham-Controlled, Double-Blind Clinical Study"

_biomedicines, 2025, doi:10.3390/biomedicines13010176_

Round 1

Reviewer 1 Report

Comments and Suggestions for Authors

This study investigates the effects of combining acupuncture (AP) with kinesitherapy (KTH) compared to KTH alone and optimal drug therapy (ODT) in improving the physical activity of patients hospitalized with acute decompensated chronic heart failure with reduced ejection fraction (HFrEF). The randomized, double-blind trial reports significant improvements in physical functionality and biomarker levels in the combined therapy group (E2), suggesting synergistic benefits of early rehabilitation using AP and KTH.

Questions:

a) Could you diminish the title of your manuscript, which should be more attractive? 

b) How were the double-blinded and randomization processes? 

c) Did you not lose any patient during follow-up? Explain it. 

d) Expand the discussion on generalizability and long-term outcomes.

e) Simplify technical language for improved readability.

f) Expand on the limitations, such as the short rehabilitation period and potential placebo effects of acupuncture.

g) Discuss the cost-effectiveness and logistical challenges of incorporating acupuncture into standard care.

h) Include practical insights on the feasibility of acupuncture integration in different healthcare settings.

Reviewer 2 Report

Comments and Suggestions for Authors

This is a very useful and informative paper. It could benefit by explanation of some technical terms which may not be familiar to readers who work outside the field of cardiology rehabilitation. 

In particular, acronyms and technical terms should be explained, and a list of abbreviations could be included (to precede the References).

In what follows, numbers indicate line numbers in the article pdf. 

24. What does ‘ 2- or 6-minute’ refer to – endurance walking, the Barthel test, or something else?

26. Define ‘NYHA’ in Abstract and main text

31. ‘in both E groups’ - but not the control group?

49. ‘I A’ – this should be 1 A, and explained for those not familiar with medical classifications, with a reference that give a clear definition

61. ‘Researchers from around the world,’ – what does this refer to? Kitzman et al. was written by many authors, but all were based in US institutions

63. Why mention the date of the German study but not that by Kitzman et al?  

75. Explain what an ‘ergo bike’ is

78. Give a brief description of the Barthel index earlier than in line 342.

100. ‘future’ – do the authors mean ‘prospective’? This paragraph is unclear – is  it a proposal for a future study, or a description of a study already carried out?

103. Location (city) of the MMA?

116. Define SGLT-2

127. Explain ‘decompensated heart failure’ v ‘compensated’

129. Provide a reference for the type of acupuncture placebo used

158. How soon after treatment was venous blood taken?

180. Describing people as ‘units’ sounds very impersonal! 38 per group – but Figure 1 shows 40 per group – please clarify

200/285. As far as this reviewer knows, SPANOVA does not exist in SPSS – please clarify!

204. Include date of Ethics Committee meeting

Figure 1: What does ‘Duplicates’ mean? 

Table 1. Does Sigma mean ‘total’?

Table 2. Consider presenting effect sizes (Kendall’s W), not just Friedman test p-values and % of change [eta2 is used in line 287]

271. Difference at baseline suggests that those in group E2 may have had an initial advantage!

273. Is it therefore strictly speaking true that ‘all groups showed similar endurance walking test results’ when admitted?

Graphs 1 and 2: Label axes. Should ‘Graphs’ be considered a ‘Figure’? If so, renumber.

Graphs 1 and 2, Figure 2. Note that colour of plot for E2 does not match that in the caption. 

322/Table 4. Median 10 days hospitalization for Group E2 – again, did this give E2 a slight advantage?

Table 4, Table 5: Use ‘5.00’, not ‘5,00’.

354. This appears to contradict 322/Table 4!

358. ‘By day 10, the E2 and E1 groups had comparable functional status’ [could use ‘status’ rather than ‘statuses’] – does this suggest AP only has short-term benefit?

401-2. What is ‘I2’?

References

484. ‘ACSM’, not ‘Acsm’ 

537. Ensure textbook author’s name is correct (surname, not given name). 

595. Ditto: authors are Zhong, Zhu and Zhong. 

Check with Biomedicines Journal whether all authors need to be included in References, or only the first few (3, 5?)

Comments on the Quality of English Language

This reviewer makes some suggestions to improve language presentation, but does not expect the authors to follow them if they disagree (numbers indicate line numbers in the article pdf).

2-6. Title is too complicated – change and simplify wording. e.g.:

Acupuncture and kinesitherapy improve physical activity more than kinesitherapy alone in patients with acute decompensated chronic heart failure with reduced ejection fraction who are already on optimal drug therapy. A randomized, sham-controlled, double-blind clinical study

17. ‘with … with’ is awkward. Try ‘suffering from’ ?

25. ‘ The 120 patients’  - Omit ‘the’; Simply use ‘120 patients’

70. ‘trends, however,’  try ‘trends; however,’

144. ‘Every individual patient had a collection of needles.’ – this is unclear – the paper already states that ‘Fresh sterile acupuncture needles were utilized.‘ Were needles re-used after sterilization, but only for the same patient? Or were the needles for single use only?

145. 'business’  - try  ‘business (working)’ for non-US readers

201-2. Reword

241. Use ‘Median’ rather than ‘Me’ – or at least explain it here, before doing so in Caption to Table 2.

259. ‘7.62’ – try ‘7.62m’

382. ‘Bruno Delgado's study’ – try ‘the 2022 study by Delgado et al.’ cf. 404 (omit ‘Neil’), 413 (omit ‘Gordon’), 415 (omit given name Hui)

Round 2

Reviewer 1 Report

Comments and Suggestions for Authors

All requests were addressed. Well done.